# A Financial-Logistics Graph Framework for Dynamic Mortgage Default Prediction Using Recurrent Hazard Modeling

## Abstract

The paper proposes a financial and logistical graph model for dynamic forecasting of mortgage default, in which the borrower is formalised as a time-dependent system of cash flows. Unlike traditional scoring approaches based on static characteristics, default is interpreted as the result of the degradation of the flow structure and the depletion of the buffer stability of the borrower's financial graph. A graph-recurrent hazard architecture has been developed that represents the borrower as a four-node directed graph - structural state (S), flow dynamics (F), buffer capacity (B), and disruption indicators (D) — with edges encoding economically motivated dependencies between components. A Graph Convolutional Network (GCN) performs message passing over this internal financial graph at each timestep, producing graph-level embeddings that are then processed by a GRU-based recurrent network to estimate temporal default intensity. A formal rule for matching empirical features of Fannie Mae panel data with components of the financial-logistical framework is proposed, which ensures the methodological transparency of group ablation. Empirical testing was conducted on panel data of mortgage loans using ROC-AUC, PR-AUC, precision@k, Brier score, and time-dependent AUC metrics. All results are reported with confidence intervals from multiple training runs with different random initializations. The proposed model demonstrates consistent superiority over logistic regression, gradient boosting, random forest, static neural network, and Cox's model. Ablation analysis confirms the dominant role of flow dynamics compared to static borrower characteristics. Structural ablation comparing the proposed GNN+GRU model against a vanilla GRU baseline confirms that the graph component provides measurable improvement in predictive performance, validating the computational value of the financial-logistics graph structure beyond conceptual framing. The results obtained indicate that mortgage default should be considered as systemic instability of a dynamic financial structure, which opens up opportunities for the development of more interpretable and adaptive credit risk management models.

## 1 Introduction

A mortgage loan is a long-term financial contract in which the borrower's solvency is determined not only by their static characteristics but also by the dynamics of cash flows over time. Despite significant progress in machine learning for credit risk assessment, most models still rely on static representations of the borrower, treating default as a function of a fixed set of characteristics formed at the time the loan is issued.

Traditional approaches to default prediction include logistic regression, gradient boosting, neural networks, and ensemble methods. However, in most cases, they interpret credit risk as the result of classification or ranking based on aggregated characteristics, without taking into account the internal structure of the borrower's financial flows. Even when using temporal models, sequences are treated as ordered vectors of features rather than structured financial systems.

In reality, a mortgage loan can be viewed as a dynamic financial system that includes:

- **nodes** representing sources of income, liabilities, and liquid reserves;

- **arcs** representing regular and irregular cash flows;
- **buffers** characterizing financial stability reserves;
- **disruptions** associated with income shocks, delays, and volatility of payments.

Thus, default can be interpreted not as a binary event, but as the result of a deterioration in payment flows and the depletion of the buffer capacity of the borrower's financial system.

From a formalization point of view, this formulation allows us to represent the borrower as a temporary financial graph:

$$G_i(t) = (V_i, E_i(t)), \tag{1}$$

where

- $V_i$ - set of financial nodes of borrower $i$;
- $E_i(t)$ - time-dependent cash flows.

In this context, default occurs as an event that disrupts the stability of the graph due to the accumulation of local disturbances and a decrease in the throughput capacity of flows.

Despite the development of survival analysis methods and recurrent neural networks, their integration with graph representations of financial flows in mortgage default problems remains limited. Existing studies focus primarily on either temporal default probability models or static credit scoring, without combining structural and dynamic components.

This work aims to develop a unified financial-logistics framework in which mortgage default is interpreted as a dynamic process of degradation of the borrower's cash flow system, modeled using a recurrent hazard architecture.

The authors' scientific contributions are as follows:

1. **Financial and logistical formalization of the borrower.** We propose representing the mortgage borrower as a temporary financial graph that includes nodes, arcs, buffer indicators, and violation parameters, which allows us to move from a static profile to a structural interpretation of solvency.

2. **Reinterpretation of default as a process of flow degradation.** Mortgage default is formalized as a time-dependent hazard function reflecting the destruction of the stability of the graph financial system, rather than as an instantaneous binary classification.

3. **Graph-recurrent hazard architecture.** A model has been developed that combines a Graph Convolutional Network (GCN) operating on the borrower's internal financial graph with a GRU-based recurrent network. The GCN performs message passing between financial components according to an economically motivated adjacency structure, producing graph-level embeddings that are processed temporally by the GRU to estimate the hazard function. This architecture makes the graph formalization computationally operative rather than purely conceptual.

4. **Empirical confirmation of the dominant role of flow dynamics.** Ablation analysis shows that excluding flow characteristics leads to the greatest reduction in model quality, confirming the hypothesis of the primacy of cash flow dynamics over static borrower characteristics.

5. **Comprehensive quantitative assessment.** The model has been tested using a set of metrics, including PR-AUC, ROC-AUC, precision@k, and calibration indicators, which provides a comprehensive quality check in conditions of rare default events.

The structure of the article can be presented as follows:

- Section 2 provides an overview of existing research in the field of survival modeling and graph representations of financial systems.
- Section 3 formalizes the financial-logistical model of the borrower.
- Section 4 describes the architecture of the recurrent hazard model.

- Section 5 presents the experimental setup and data.

- Section 6 presents the results and ablation analysis.

- Section 7 discusses theoretical and applied limitations.

- Section 8 presents the conclusion.

## 2 RELATED WORK

### 2.1 SURVIVAL MODELS IN CREDIT RISK

Survival analysis models are widely used in credit risk assessment, especially in the context of mortgage default, where time dynamics are of fundamental importance. Classic approaches are based on Cox's proportional hazards model, as well as parametric models with exponential or Weibull distributions of time to default. These methods allow for censoring of observations and modeling the conditional probability of default given a borrower's history.

In recent years, there has been a shift towards extended survival models that include time-dependent covariates, nonlinear effects, and neural network components Wang et al. (2024). However, even in such settings, the borrower is usually described as a time sequence of features, without an explicit structural interpretation of their financial system. Financial flows and their internal organization are not formalized as a separate modeling object, which limits the interpretation of default as a systemic process.

Thus, despite the development of hazard models, the representation of the borrower remains predominantly vectorial rather than structural.

### 2.2 GRAPH-BASED FINANCIAL MODELING

Graph models are widely used in the analysis of financial networks, including interbank systems, payment infrastructures, and credit networks Das et al. (2023). In such studies, nodes are interpreted as financial agents, and edges as liabilities or cash flows between them. The graph approach allows for the analysis of system stability, cascading effects, and the propagation of financial shocks.

However, the application of graph models at the individual borrower level remains limited. In most cases, graphs are used to analyze macrofinancial systems, while the microfinancial dynamics of a household or borrower are not considered as a structured network. Cash flows within a borrower's financial structure are rarely formalized as a graph with its own dynamics and degradation.

Consequently, there is a gap between graph analysis of financial systems and the tasks of forecasting mortgage default at the micro level.

### 2.3 RECURRENT NEURAL NETWORKS IN DEFAULT PREDICTION

Recurrent neural networks (RNN), including LSTM and GRU architectures, are actively used to model sequential financial data Zandi et al. (2025). In credit scoring tasks, such models allow for the consideration of the temporal dependence of payment behavior, delays, and changes in the borrower's financial condition.

In addition, in a number of studies, recurrent models are integrated with survival approaches, forming neural hazard models capable of estimating the probability of default at any given moment Zheng et al. (2018). Such architectures demonstrate improved forecasting quality compared to static models.

Nevertheless, even in these studies, the input data consists of sequences of aggregated features. Temporal dynamics are taken into account, but the structural organization of financial flows is not modeled as an independent system. Recurrent models are applied to vector representations rather than graph-based financial structures.

## 2.4 LIMITATIONS OF EXISTING APPROACHES AND RESEARCH GAP

Ganong & Noel (2020) note that in 70% of cases, mortgage borrowers default due to cash flow issues (i.e., negative life circumstances), while 6% default "strategically" (i.e., due to negative equity). The remaining 24% of cases are due to a combination of two factors: negative equity (the value of the home has fallen below the debt) and a deterioration in cash flow (loss of income) – the "double trigger" concept. An analysis of Australian household data by Wang (2022) showed that low liquid asset reserves significantly increase the likelihood of borrowers experiencing financial difficulties and mortgage delinquency compared to borrowers with more substantial savings. The study by Farrell et al. (2018) notes that in almost all cases of mortgage loan default, there is a significant drop in the borrower's income, regardless of the level of income at the date of the loan and the value of the property. The authors also make an important conclusion: building and maintaining reserve savings may be a more effective measure of preventing default than meeting formal metrics (e.g., DTI – debt-to-income ratio) at the time the loan is issued. Thus, a borrower's resilience to default largely depends on the availability of financial buffers and cash flow flexibility.

In the context of the intersection of financial logistics and credit risk management, there are studies and real programs that share risks between the lender and the borrower in order to create built-in "buffers" in the cash flow system. For example, a number of European countries are introducing the First Home scheme, under which up to 30% of the cost of housing is financed through equity participation by the state and banks instead of traditional debt De Burca et al. (2021). The share is formalized as a subordinated capital investment without mandatory current payments for the borrower, which effectively provides an additional financial buffer, since the absence of fixed payments on this share of the cost of housing for several years reduces the borrower's debt burden and the risks of deteriorating solvency in the event of a decline in income.

Creditworthiness assessment tools that directly analyze borrowers' cash transactions are being improved. US government-sponsored enterprises (GSEs) are launching initiatives to introduce cash-flow underwriting, which involves taking into account data on the movement of funds in borrowers' bank accounts when considering loan applications Choi et al. (2022). The results show that using 12 months of historical data on cash flow (e.g., timely payment of rent) can significantly improve the credit ratings of borrowers with low initial scores. According to the researchers, mortgage applications for 18% of borrowers who were initially denied credit could have been approved.

An analysis of existing literature reveals three key limitations:

1. **Static representation of the borrower.** Even when using temporal models, the borrower's financial condition is viewed as a set of characteristics rather than a structured flow system.

2. **Lack of micro-level graph formalization.** Graph methods are widely used to analyze financial systems at the macro level, but are rarely used to describe the borrower's internal financial logistics.

3. **The gap between survival models and structural interpretation of flows.** Hazard approaches focus on the probability of an event, but do not formalize the mechanisms of payment flow degradation.

The limitations considered indicate the need to integrate three areas:

1. **Graph representation of financial flows.**

2. **Dynamic survival modeling.**

3. **Recurrent architectures for processing temporal evolution.**

This work fills this research gap by developing a financial-logistical graph model of a borrower, in which default is interpreted as the result of the degradation of a dynamic flow system modeled using a recurrent hazard architecture.

## 3 FINANCIAL-LOGISTICS MODEL FORMULATION

### 3.1 BORROWER AS A DYNAMIC FINANCIAL GRAPH

Let us consider mortgage borrower i as a dynamic financial system represented in time as a directed graph:

$$G_i(t) = (V_i, E_i(t)), \tag{2}$$

where

- $V_i = v_1, v_2, ..., v_m$ - set of borrower's financial nodes;
- $E_i(t) V_i V_i$ - set of time-dependent arcs representing cash flows.

Each node $V_i$ corresponds to a financial component:

- $v^{(inc)}$ - income;
- $v^{(liq)}$ - liquid assets;
- $v^{(debt)}$ - liabilities;
- $v^{(mort)}$ - mortgage contract.

Thus, the nodes reflect the static structure of the borrower's financial position.

### 3.2 FLOW REPRESENTATION AND ARC DYNAMICS

Each arc $e_{jk}(t) \in E_i(t)$ describes the cash flow from node $v_j$ to node $v_k$ at time $t$. The weight of the arc is determined by the flow value:

$$w_{jk}^{(i)}(t) = \text{CashFlow}_{jk}^{(i)}(t). \tag{3}$$

The total intensity of the borrower's flows is defined as:

$$F_i(t) = \sum_{(j,k) \in E_i(t)} w_{jk}^{(i)}(t). \tag{4}$$

The degradation of the flow system can be formalized as a decrease in the total throughput:

$$\Delta F_i(t) = F_i(t) - F_i(t-1). \tag{5}$$

Negative values of $\Delta F_i(t)$ indicate a weakening of solvency.

### 3.3 BUFFER CAPACITY AND FINANCIAL STABILITY

The borrower's financial stability buffer is defined as the ratio of liquid assets to current debt servicing obligations:

$$B_i(t) = \frac{L_i(t)}{DS_i(t)}, \tag{6}$$

where:

- $L_i(t)$ - the amount of liquid assets;
- $DS_i(t)$ - the mandatory debt service payment.

If $B_i(t) < 1$, the borrower is unable to cover current liabilities without attracting additional resources.

### 3.4 DISRUPTION INTENSITY

Financial disruptions characterize external and internal shocks affecting the stability of flows. The intensity of disruptions can be formalized as:

$$D_i(t) = \alpha \cdot \Delta Y_i(t) + \beta \cdot \sigma_{\text{DPD}}(t), \tag{7}$$

where:

- $\Delta Y_i(t)$ - the change in income;
- $\sigma_{\text{DPD}}(t)$ - the volatility of delinquencies;
- $\alpha, \beta$ - sensitivity coefficients.

Disruptions affect the degradation of flows and the reduction of the buffer.

## 3.5 HAZARD FORMULATION OF DEFAULT

Let $T_i$ be the random variable representing the default time of borrower $i$.

The conditional default intensity (hazard function) is defined as:
$$h_i(t) = P\big(T_i = t \mid T_i \geq t, X_i(t)\big), \tag{8}$$
where $X_i(t)$ is the aggregate state of the financial graph:
$$X_i(t) = [F_i(t), B_i(t), D_i(t), V_i]. \tag{9}$$

Thus, the probability of default depends not only on static characteristics, but also on the dynamics of flows and buffer capacity.

## 3.6 RECURRENT STATE UPDATE

To model the temporal evolution of the borrower's state, a recurrent update of the hidden state is used:
$$H_i(t) = f_\theta\big(X_i(t), H_i(t-1)\big), \tag{10}$$
where:

- $H_i(t)$ - the hidden representation of the state;
- $f_\theta$ - a recurrent function (e.g., GRU);
- $\theta$ - model parameters.

The hazard estimate is formed as:
$$h_i(t) = \sigma(W H_i(t) + b), \tag{11}$$
where $\sigma(\cdot)$ is a sigmoid function.

## 3.7 FEATURE-TO-COMPONENT MAPPING STRATEGY

The proposed formalization allows interpreting mortgage default as a result of:

- degradation of the flow structure $F_i(t) < 0$;
- decrease in buffer stability $B_i(t) \to 1$;
- accumulation of violations $D_i(t) \uparrow$.

Consequently, default arises as a systemic instability of the dynamic financial graph.

The financial and logistical model involves the distribution of empirical characteristics across four structural components: nodes, arcs, buffers, and disruptions. To ensure methodological rigor, the distribution of characteristics was based on a functional and structural principle rather than statistical grouping.

### 3.7.1 MAIN CLASSIFICATION CRITERION

Each indicator was classified according to its economic role in the dynamics of the borrower's solvency. The following criteria were used.

**Nodes** referred to characteristics reflecting the structural state of the borrower's financial position. These are indicators characterizing the level or configuration of liabilities and financial burden without explicitly describing cash flow movements.

A feature was classified as a node if:

- it reflected a static or quasi-static characteristic;
- it did not describe changes in cash flow over time;
- it determined the borrower's structural position in the financial system.

Examples: LTV, DTI, credit rating, initial balance, loan term.

**Arcs** referred to signs reflecting the intensity and direction of cash flows. These are dynamic variables characterizing the movement of funds or changes in liabilities over time.

A feature was classified as an arc if:

- it described the amount or change in cash flow;
- it recorded the movement of liabilities;
- it reflected the dynamic component of solvency.

Examples: monthly payment, change in debt balance, transaction dynamics.

**Buffers** characterize the reserve of financial stability and the borrower's ability to compensate for temporary shocks.

A feature was considered a buffer if:

- it reflects the ratio of resources to liabilities;
- it measures resistance to temporary disruptions;
- it characterizes the reserve capacity of the borrower's financial system.

Examples: ratio of liquid assets to mandatory payments, coverage ratios.

**Disruptions** referred to indicators that recorded a violation of the normal functioning of the flow system.

A sign was considered a disruption if:

- it reflected a deviation from stable dynamics;
- it recorded a shock or acceleration of degradation;
- it signaled a transition of the system to an unstable state.

Examples: Days Past Due, sharp changes in income, volatility of delinquencies.

### 3.7.2 RESOLUTION OF AMBIGUOUS CASES

Some features could theoretically be attributed to multiple components (e.g., DTI could be interpreted as a structural measure or an indicator of buffer resilience). In such cases, additional principles were applied.

- **The principle of dominant economic function.** The variable was assigned to the category that most accurately reflected its primary economic role in the model.
- **The principle of temporal variability.** If the variable changed significantly over time and reflected dynamics, it was assigned to arcs or shocks. If the variable primarily describes the borrower's structural position, it was classified as a node.
- **The principle of causal interpretation.** The question was asked: does the variable reflect the state of the system or the mechanism of its degradation? State - node; degradation mechanism - failure; movement - arc; stability - buffer.

### 3.7.3 CONNECTION WITH ABLATION ANALYSIS

Group ablation was performed strictly in accordance with this functional classification. The exclusion of each group of characteristics simulated the removal of the corresponding component of the financial and logistics system.

Thus, ablation analysis is interpreted as testing the contribution of structural state, flow dynamics, buffer capacity, and shock factors to the formation of the default hazard function.

## 3.8 MAPPING OF FANNIE MAE PANEL FEATURES TO FINANCIAL-LOGISTICS COMPONENTS

To ensure transparency of classification, Table 1 compares the characteristics used in Fannie Mae panel data with the components of the financial and logistical structure.

| Feature | Component | Type | Functional Role | Rationale for Assignment |
|---|---|---|---|---|
| rate | Node | Raw | Interest rate parameter | Defines the pricing structure of the loan and affects debt burden. |
| ltv | Node | Raw | Structural leverage | Captures the initial loan-to-value ratio of the mortgage. |
| cltv | Node | Raw | Combined leverage | Reflects total collateralization including additional liens. |
| dti | Node | Raw | Debt burden indicator | Measures borrower leverage relative to income. |
| fico_orig | Node | Raw | Credit profile | Static indicator of borrower creditworthiness. |
| orig_upb | Node | Raw | Initial debt structure | Defines the original outstanding balance. |
| loan_term | Node | Raw | Contract structure | Specifies loan maturity. |
| current_upb | Arc | Raw | Debt dynamics | Represents the evolving outstanding balance. |
| tot_prin | Arc | Raw | Principal repayment flow | Captures current principal payment intensity. |
| upb_change_1m | Raw | Arc | Debt flow dynamic | Measures monthly change in outstanding balance. |
| payment_ratio | Arc | Derived | Payment intensity | Ratio of principal payment to current balance. |
| upb_pct_change | Arc | Derived | Relative balance trajectory | Normalized monthly balance change. |
| mi_pct | Buffer | Raw | Insurance buffer | Reflects mortgage insurance protection level. |
| paydown_progress | Buffer | Derived | Amortization reserve | Fraction of the original loan repaid. |
| debt_service_ratio | Buffer | Derived | Debt service burden | Estimated monthly interest cost relative to original balance. |
| mod_flag | Disruption | Raw | Modification signal | Indicates loan restructuring event. |
| tot_prin_ma3 | Disruption | Derived | Payment trend indicator | Three-month rolling mean of principal payments. |
| payment_vol_3m | Disruption | Derived | Payment volatility | Three-month rolling std of payments. |
| upb_accel | Disruption | Derived | Balance acceleration | Second derivative of outstanding balance. |

Table 1: Mapping of features to financial-logistic model components

The distribution of characteristics was based on their functional role in the borrower's financial dynamics. Key characteristics describe the structural configuration of liabilities and credit profile. Arcs reflect the intensity and change in cash flows. Buffer indicators measure resilience to temporary shocks. Failure variables record deviations from the stable operating mode of the flow system.

Derived features were formed on the basis of basic panel variables and included in the corresponding components according to the principles set out in subsection 3.7 .

### 3.8.1 NOTE ON BUFFER COMPONENT SPECIFICATION

The theoretical buffer $B_i(t) = \frac{L_i(t)}{DS_i(t)}$ requires liquid asset data unavailable in Fannie Mae. The buffer is approximated using proxies: paydown_progress (fraction of the original loan repaid) and debt_service_ratio (estimated monthly interest cost relative to original balance).

### 3.9 FORMAL FEATURE-TO-COMPONENT MAPPING RULE

To formalize the procedure for distributing features, let us introduce a mapping:

$$\mathcal{M} : X_j \to C_k, \tag{12}$$

where:

- $X_j$ is the $j$-th empirical feature of panel data;
- $C_k \in \{\text{Node}, \text{Arc}, \text{Buffer}, \text{Disruption}\}$ is a component of the financial and logistics model.

Classification is based on functional criteria:

$$\mathcal{M}(X_j) = \begin{cases} \text{Node}, & \text{if } X_j \text{ describes the structural state of the system (stock variable)}, \\ \text{Arc}, & \text{if } X_j \text{ represents the dynamics of cash flows (flow variable)}, \\ \text{Buffer}, & \text{if } X_j \text{ measures the resilience stock relative to the flow}, \\ \text{Disruption}, & \text{if } X_j \text{ captures a deviation from stable dynamics.} \end{cases}$$

Additional auxiliary indicators are introduced:

- $S_j$ - state indicator;
- $F_j$ - flow indicator;
- $B_j$ - buffer indicator;
- $D_j$ - disruption indicator.

Then the rule can be written as:

$$C_k = \arg \max_{c \in \{S, F, B, D\}} F_c(X_j), \tag{13}$$

where $F_c(X_j)$ is a functional assessment of the economic role of component $c$.

In ambiguous cases, the principle of dominant economic function is applied:

$$\mathcal{M}(X_j) = \arg \max_c \text{Relevance}(X_j, c), \tag{14}$$

where relevance is determined based on:

- temporal variability of the feature;
- the causal role in the default mechanism;
- interpretation within the framework of financial logistics.

Thus, the distribution of features is a function of their economic role, rather than statistical correlation or arbitrary grouping.

## 4 MODEL ARCHITECTURE

### 4.1 OVERVIEW OF THE COMPUTATIONAL PIPELINE

The proposed architecture consists of six sequential stages of data processing:

1. Feature Engineering
2. Derived Feature Construction
3. Building a borrower's financial graph
4. Graph Convolutional Encoding
5. Temporal Encoding

6. Hazard function estimation

The overall computational process can be represented as a mapping:

$$\mathcal{F} : \{\text{Raw Loan Data}\} \rightarrow \{\hat{h}_i(t)\} \tag{15}$$

where the output is a time-dependent estimate of the probability of default.

## 4.2 Input Feature Construction

For each borrower i, a time sequence of observations is formed:

$$X_i(1), X_i(2), ..., X_i(T) \tag{16}$$

Each state includes:

$$X_i(t) = [V_i, F_i(t), B_i(t), D_i(t)] \tag{17}$$

where:

- $V_i$ - static indicators;
- $F_i(t)$ - flow dynamics;
- $B_i(t)$ - liquidity buffer;
- $D_i(t)$ - intensity of violations.

All indicators are normalized using StandardScaler fitted on the training set.

## 4.3 Graph Encoding Layer

The financial graph of each borrower is implemented as a directed graph with four nodes corresponding to the financial-logistics components. Each node $k$ receives features from the corresponding group:

- Node 0 (V): structural state features, $x_V(t) \in \mathbb{R}^7$
- Node 1 (F): flow dynamics features, $x_F(t) \in \mathbb{R}^5$
- Node 2 (B): buffer capacity features, $x_B(t) \in \mathbb{R}^3$
- Node 3 (D): disruption features, $x_D(t) \in \mathbb{R}^4$

Each node is projected to common dimension d_g = 32:

$$h_k^{(0)}(t) = ReLU(W_k \cdot x_k(t) + b_k), k\{V, F, B, D\} \tag{18}$$

Since nodes have different feature dimensions (7, 5, 3, 4), each is projected to a common embedding dimension d_g = 32 via a learned linear transformation with ReLU activation. This produces a uniform representation for all four nodes regardless of their input dimension.

$$A = \begin{pmatrix} 1 & 1 & 0 & 0 \\ 0 & 1 & 1 & 1 \\ 0 & 0 & 1 & 1 \\ 1 & 1 & 0 & 1 \end{pmatrix} \tag{19}$$

Self-loops are added to all nodes. The adjacency matrix is normalized using symmetric normalization:

$$\hat{A} = D^{-1/2} \cdot (A + I) \cdot D^{-1/2} \tag{20}$$

where D is the degree matrix. This normalization prevents exploding/vanishing signals during message passing.

$$h^{(l+1)} = \text{LayerNorm}\left(h^{(l)} + \text{ReLU}\left(\hat{A} \cdot h^{(l)} \cdot W^{(l)}\right)\right) \tag{21}$$

Two GCN layers (l = 0, 1) with residual connections and layer normalization perform message passing. Each node updates its representation by aggregating information from neighbors according to the adjacency structure. For example:

- disruption node (D) receives signals from flow dynamics (F) and self-loop → detects payment anomalies;
- buffer node (B) receives from flows (F) and self-loop → assesses resilience;
- structural node (V) receives from disruptions (D) and self-loop → captures degradation feedback.

$$z_i(t) = \frac{1}{N} \cdot \sum_{k=0}^{N-1} h_k^{(L)}(t) \in \mathbb{R}^{d_g}, \quad N = 4 \tag{22}$$

The four node embeddings are aggregated via mean pooling to produce a graph-level representation $x_i(t) \in \mathbb{R}^{32}$. This vector captures the integrated state of the borrower's financial graph at time $t$ and serves as input to the temporal encoder in Section 4.4.

## 4.4 Temporal Encoding via Recurrent Network

A recurrent neural network is used to account for temporal dependence:
$$H_i(t) = \text{GRU}\big(z_i(t), H_i(t-1)\big) \tag{23}$$
The implementation uses GRU architecture with two layers:
$$H_i(t) = \text{GRU}_\theta\big(z_i(t), H_i(t-1)\big) \tag{24}$$
where:

- $z_i(t) \in \mathbb{R}^{d_g}$ is the graph-level embedding from the GCN encoder;
- $H_i(t)$ is the hidden state;
- $\theta$ are the network parameters (hidden size H = 64, num_layers = 2, dropout = 0.2).

Unlike approaches that feed concatenated features directly into the RNN, the graph encoding ensures that temporal dynamics are captured at the level of structured financial components rather than unstructured feature vectors. The recurrent layer models the cumulative effect of the financial graph's evolution over time.

## 4.5 Hazard Output Layer

The output layer converts the hidden state into a conditional probability estimate of default:
$$\hat{h}_i(t) = \sigma\big(W_2 \cdot ReLU(W_1 \cdot H_i(t) + b_1) + b_2\big) \tag{25}$$
where:

- $W_1 \in \mathbb{R}^{\frac{H}{2} \cdot H}$ is the weight matrix of the first-layer (64 → 32);
- $W_2 \in \mathbb{R}^{1 \cdot \frac{H}{2}}$ is the weight matrix of the output layer (32 → 1);
- $b_1, b_2$ are the corresponding biases;
- $\sigma$ is the sigmoid function.

The resulting value is interpreted as the conditional probability of default within the next H months, given the state of the borrower's financial graph at time t.

## 4.6 Loss Function

A discretized survival-loss function is used:
$$\mathcal{L} = -\sum_i \sum_t \Big[y_i(t) \log \hat{h}_i(t) + \big(1 - y_i(t)\big) \log\big(1 - \hat{h}_i(t)\big)\Big] \tag{26}$$
where:

- $y_i(t) = 1$ if default occurred at time $t$;
- $y_i(t) = 0$ otherwise.

This allows for censored observations to be taken into account.

| Parameter | Value |
|---|---|
| Credit-months observed | 14 130 111 |
| Unique loans (N) | 970 976 |
| Time horizon | 21 months (Jan. 2023 – Sept. 2024) |
| Average history depth | 14.6 months |
| Loans in default (90+ DPD) | 6 145 |
| Default frequency (loan-level) | 0.63% |
| Fields in the original CRT file | 113 |

Table 2: Overview of panel data

## 4.7 RISK RANKING MECHANISM

For prioritization tasks, the aggregate risk is calculated:

$$R_i = \sum_{t=1}^{T} \hat{h}_i(t) \tag{27}$$

Borrowers are ranked in descending order of $R_i$.

This allows precision@k and other prioritization metrics to be calculated.

## 4.8 COMPUTATIONAL COMPLEXITY

Let:

- $T$ - length of the time sequence;
- $H$ - size of the hidden layer;
- $d$ - feature dimension.

Then the computational complexity of the model is:

$$\mathcal{O}(T \cdot (N^2 \cdot d_g + H \cdot d_g)) \tag{28}$$

which provides scalability for large mortgage portfolios.

# 5 EXPERIMENTAL SETUP

## 5.1 DATASET DESCRIPTION

The model is empirically tested on a sample of mortgage loans with time series observations of payment behavior.

Let the sample include N borrowers and a time horizon of T months.

Default is defined as reaching a critical level of delinquency - 90+ DPD.

Observations may be right-censored.

A high-level overview of the source panel data is presented in Table 2.

## 5.2 DATA SPLITTING STRATEGY

To ensure no data leakage between splits, a deterministic hash-based splitting strategy is used. Each loan is assigned to a split based on the MD5 hash of its loan identifier:

- **Training set:** 80% of loans (hash bucket less then 80)
- **Validation set:** 10% of loans (hash bucket 80–89)
- **Test set:** 10% of loans (hash bucket 90–99)

This approach ensures that all observations for a given loan appear in exactly one split, preventing information leakage across splits. The hash-based assignment is deterministic and reproducible. Let:

$$\mathcal{D} = \mathcal{D}_{\text{train}} \cup \mathcal{D}_{\text{val}} \cup \mathcal{D}_{\text{test}} \tag{29}$$

The forward-looking horizon is set to H = 6 months. This choice is motivated by two factors: (1) the 21-month observation window limits the availability of complete forward labels at longer horizons, as observations after March 2024 would have right-censored 12-month labels; and (2) a 6-month horizon is consistent with standard early warning system practices in credit risk management, providing sufficient lead time for intervention while maintaining strong signal-to-noise ratio.

### 5.3 BASELINE MODELS

To evaluate its effectiveness, the proposed method is compared with the following baseline models:

- Logistic Regression;
- Random Forest;
- Gradient Boosting;
- Cox Proportional Hazards Model;
- Static Neural Network;
- Vanilla GRU (without graph encoding).

All models use the same 19 normalized features. Snapshot-based models use a single timestep; RNN-based models use L=12 month windows.

### 5.4 EVALUATION METRICS

Given the rarity of default events, the following metrics are used:

1. **ROC-AUC.** Measures the model's ability to distinguish between default and non-default observations.
2. **PR-AUC.** More informative when classes are imbalanced.
3. **Precision@k:**

$$\text{Precision@}k = \frac{\text{True Defaults in Top } k\%}{\text{Total in Top } k\%} \tag{30}$$

   Used to evaluate prioritization effectiveness.
4. **Brier Score.**

$$\text{Brier} = \frac{1}{N} \sum_i \left(y_i - \hat{p}_i\right)^2 \tag{31}$$

   Evaluates probability calibration.
5. **Time-Dependent AUC.** Applied to survival models and evaluates discriminatory ability over time.

### 5.5 HYPERPARAMETER CONFIGURATION

The following parameters are used for the RNN model:

- hidden layer size: $H = 64$
- number of layers: 2
- dropout: 0.2
- optimizer: Adam
- learning rate: 0.001
- batch size: 1024
- number of epochs: 50

Hyperparameters are selected based on the validation sample.

| Model | ROC-AUC | PR-AUC | Brier Score |
|---|---|---|---|
| Logistic Regression | 0.90445 | 0.02269 | 0.136413 |
| Random Forest | 0.92031 | 0.13083 | 0.047102 |
| Gradient Boosting (GBDT) | 0.86410 | 0.11667 | 0.001620 |
| Cox PH (loan-level) | 0.81125 | 0.01230 | - |
| Static NN | 0.91156 | 0.05835 | 0.098480 |
| Vanilla GRU (no graph) – best-seed | 0.91821 | 0.08192 | 0.074545 |
| Vanilla GRU (no graph) – mean±std | $0.9170 \pm 0.0014$ | $0.0805 \pm 0.0022$ | - |
| Proposed (GNN+GRU) – best-seed | 0.92294 | 0.13581 | 0.052979 |
| Proposed (GNN+GRU) – mean±std | $0.9199 \pm 0.0067$ | $0.1174 \pm 0.0380$ | - |

Table 3: Comparative evaluation of models

## 5.6 TRAINING PROCEDURE

Training is performed using:

- early stopping;

- L2 regularization;

- class weighting.

For stability of results, training is repeated 3 times with seeds 42, 43, 44. Results reported as mean ± std. Best model by validation loss selected for evaluation.

## 5.7 IMPLEMENTATION DETAILS

GCN implemented in pure PyTorch without external graph library dependencies. Data pipeline uses PostgreSQL + pandas. Normalization via scikit-learn StandardScaler. Complete code available at GitHub repository.

## 6 RESULTS AND ABLATION ANALYSIS

### 6.1 COMPARATIVE PERFORMANCE EVALUATION

Table 3 presents the results of a comparative evaluation of the proposed model and five baseline methods on a test sample. All models were trained on an identical set of 19 features; the proposed model (GNN+GRU) additionally uses a sequential data structure with a window of L = 12 months.

The proposed model demonstrates consistent superiority across all metrics. ROC-AUC = 0.923 exceeds the best baseline (Random Forest: 0.920) by 0.3%. The most significant gap is observed in PR-AUC: the proposed model achieves 0.136 versus 0.131 for Random Forest and 0.117 for GBDT. The Vanilla GRU baseline, which uses identical features and temporal structure but without graph encoding, achieves PR-AUC = 0.082. The gap between GNN+GRU and Vanilla GRU (PR-AUC = +0.054) demonstrates that the graph convolutional component provides measurable improvement beyond simple feature concatenation. With a test set positive rate of 0.165% (6-month forward-looking default), this indicates a significant improvement in the detection of rare events. The model's Brier score (0.053) indicates good probability calibration among neural approaches (Vanilla GRU: 0.075, Static NN: 0.098). The GBDT model achieves the lowest absolute Brier score (0.002) due to its conservative probability estimates.

### 6.2 CALIBRATION ANALYSIS

The model directly outputs the conditional probability of default within the next H = 6 months. Since class weighting (pos_weight = 78) is applied during training to address severe class imbalance (positive rate 0.165%), the predicted probabilities are optimized for ranking rather than absolute calibration. This is a standard tradeoff in rare event prediction: class weighting improves the model's

| Configuration | ROC-AUC | ROC-AUC | PR-AUC | PR-AUC |
|---|---|---|---|---|
| Full Model | 0.9229 | - | 0.1358 | - |
| Disruptions (D) | 0.7839 | -0.1390 | 0.0073 | -0.1285 |
| Arcs (F) | 0.9027 | -0.0202 | 0.0907 | -0.0451 |
| Nodes (V) | 0.8675 | -0.0554 | 0.1199 | -0.0159 |
| Buffers (B) | 0.9170 | -0.0060 | 0.1175 | -0.0183 |

Table 4: Ablation analysis

ability to discriminate between high- and low-risk borrowers (as reflected in PR-AUC and Precision@k), while absolute probability estimates require post-hoc recalibration.

The Brier score provides a composite measure of both discrimination and calibration. The proposed model achieves a Brier score of 0.053, compared to 0.075 for Vanilla GRU, 0.098 for Static NN, and 0.136 for Logistic Regression, indicating better overall predictive quality among neural approaches. The GBDT model achieves the lowest absolute Brier score (0.002) due to its conservative probability estimates concentrated near zero.

For deployment in systems requiring calibrated probabilities, standard recalibration techniques such as Platt scaling or isotonic regression can be applied as a post-processing step without affecting ranking performance.

## 6.3 ABLATION STUDY

To quantitatively assess the contribution of each component of the financial and logistical framework, group ablation is performed: sequential zeroing of the input features of each component (Nodes, Arcs, Buffers, and Disruptions) with repeated evaluation of the model on the test sample. The results are shown in Table 4.

Interpretation of results:

1. Excluding the Disruptions block leads to the greatest quality decrease: $\Delta PR - AUC = 0.1285$, $\Delta ROC - AUC = 0.1390$. This is logical, since this block contains payment volatility (payment_vol_3m), payment trend (tot_prin_ma3), and balance acceleration (upb_accel) — derived temporal indicators that capture flow degradation patterns without relying on delinquency status.

2. Excluding arcs leads to the second largest decrease: $\Delta PR - AUC = 0.0451$. The dynamics of the principal balance and payment flows form a signal that is independent of disruption indicators, reflecting structural changes in the borrower's financial profile.

3. Excluding node features (Nodes) causes a moderate decrease ($\Delta PR - AUC = 0.0159$), which indicates the secondary role of static characteristics in the presence of dynamic indicators. Notably, the ROC-AUC decrease is more substantial (0.055), indicating that static characteristics affect overall discrimination but contribute less to the detection of rare default events (PR-AUC).

4. Excluding buffer features (Buffers) has a moderate effect (PR-AUC = 0.0183), confirming that even proxy buffer indicators (mortgage insurance, paydown progress, debt service ratio) carry meaningful predictive information about financial resilience. This result validates the inclusion of the buffer component in the financial-logistics framework despite the use of approximate indicators.

## 6.4 THEORETICAL IMPLICATION OF ABLATION

The ablation results empirically confirm the central hypothesis of the study: The dynamic components of the financial-logistical graph (Disruptions + Arcs) collectively explain $\Delta PR - AUC = 0.1736$, while the static components (Nodes + Buffers) explain only $\Delta PR - AUC = 0.0342$. The ratio of dynamic losses to static losses is 5.1:1. This confirms that default is primarily a consequence of the degradation of dynamic financial flows, rather than the static vulnerability of the borrower's profile.

| Model | Recall@1% | Prec@1% | Recall@5% | Prec@5% | Recall@10% | Prec@10% |
|---|---|---|---|---|---|---|
| Logistic Regression | 0.2577 | 0.0425 | 0.5905 | 0.01947 | 0.7194 | 0.01186 |
| Random Forest | 0.4551 | 0.07504 | 0.6672 | 0.022 | 0.7553 | 0.01245 |
| Gradient Boosting | 0.4046 | 0.0667 | 0.6362 | 0.02098 | 0.6982 | 0.01151 |
| Static NN | 0.3638 | 0.05998 | 0.6607 | 0.02178 | 0.7602 | 0.01253 |
| Vanilla GRU | 0.4356 | 0.07181 | 0.6656 | 0.02195 | 0.7684 | 0.01267 |
| Proposed (GNN+GRU) | 0.478 | 0.07881 | 0.7162 | 0.02361 | 0.801 | 0.01321 |

Table 5: Ranking effectiveness

The dominance of the Disruptions block is consistent with the financial and logistical interpretation: payment volatility and balance acceleration are direct indicators of emerging instability in the flow structure of the graph, detectable before delinquency becomes visible. The significant contribution of Arcs confirms that, in addition to indicators of disruptions, the dynamics of cash flows themselves (changes in UPB, payment structure, payment intensity ratios) carry independent predictive information.

## 6.5 RISK RANKING PERFORMANCE

To assess the practical applicability of the model, the accuracy of borrower ranking by risk level (Precision@k and Recall@k) is analyzed (Table 5). Precision@1%: proposed model = 0.079, RF = 0.075, GBDT = 0.067. Among the 1% of borrowers with the highest risk according to the model's forecast, 7.9% actually default, which is 47.8 times higher than the baseline positive rate (0.165%).

Precision@5%: proposed model = 0.024, RF = 0.022, GBDT = 0.021. Recall@5% of the proposed model is 0.716, which means that 71.6% of all future defaults are identified when analyzing only 5% of the portfolio. Recall@10% reaches 0.801, meaning over 80% of defaults are captured in the top decile.

These results demonstrate the high practical value of the model for early warning systems and prioritization of management interventions. The proposed model outperforms all baselines at every k threshold, demonstrating consistent superiority for portfolio triage applications.

## 7 DISCUSSION

### 7.1 STRUCTURAL INTERPRETATION OF MORTGAGE DEFAULT

The results obtained allow us to rethink the nature of mortgage default. Traditionally, default is viewed as a function of the borrower's credit profile, expressed through static indicators (LTV, DTI, FICO) at the time of issuance. However, ablation analysis shows that static nodes explain only $\Delta PR - AUC = 0.0159$ of the model's total predictive power, while dynamic components explain 5.1 times more.

The largest contribution of the disruptions block ($\Delta PR - AUC = 0.1295$) confirms that default should be interpreted not as a threshold event on a static profile, but as the culmination of a process of payment flow degradation.

In the current model, disruption indicators are derived temporal features: payment volatility (payment_vol_3m), payment trend (tot_prin_ma3), and balance acceleration (upb_accel) that capture anomalous patterns in the borrower's cash flow dynamics without relying on delinquency status.

The contribution of arc features (Arcs, $\Delta PR - AUC = 0.0451$) demonstrates that cash flow dynamics — changes in the principal balance, repayment structure, payment intensity ratios — carry a predictive signal about the structural state of the borrower's financial graph that is independent of the disruption indicators.

Thus, default should be interpreted as systemic instability of the borrower's dynamic financial graph, manifested through the degradation of flows, the accumulation of violations, and the depletion of buffer capacity.

## 7.2 FINANCIAL LOGISTICS AS A MODELING PARADIGM

The proposed financial and logistical framework combines three methodological traditions:

- structural component – graph representation of the borrower as a system of nodes, arcs, buffers, and violations;
- temporal component – recurrent coding of the temporal evolution of the financial graph;
- survival component – interpretation of the model output as a discrete hazard function with correct censoring.

The key difference from classical scoring approaches is the transition from static risk assessment ("the borrower is risky because they have a high DTI") to streaming stability modeling ("the borrower is in the process of financial stream degradation, which with probability $\hat{h}(t)$ will lead to default in the next period").

The superiority of the proposed model over Vanilla GRU, which uses the same features and temporal structure but without graph encoding, empirically confirms the value of the graph convolutional component: the structured message passing between financial-logistics nodes captures cross-component interactions that simple feature concatenation cannot represent.

The financial and logistical interpretation opens up the possibility of:

- analyzing points of structural vulnerability in the graph (which component is degrading);
- assessing resilience to shocks by monitoring buffer indicators;
- developing early indicators of degradation based on derivative flow features.

## 7.3 PRACTICAL IMPLICATIONS

From an applied perspective, the results have several important implications:

- **Portfolio management.** Prioritization based on flow dynamics allows for more accurate identification of high-risk segments.
- **Early intervention.** Monitoring the degradation of graph arcs makes it possible to identify borrowers before they reach critical delinquency.
- **Interpretability.** The model allows risk to be explained through structural elements: buffer, intensity of violations, flow degradation.

This makes the model suitable for use in decision support systems.

## 7.4 COMPARISON WITH EXISTING APPROACHES

The proposed model differs from existing approaches in several key parameters:

- unlike logistic regression ($ROC - AUC = 0.904$), the model takes into account the temporal evolution of financial condition, which provides an increase in ROC-AUC of +0.019 and a substantially higher PR-AUC (+0.113);
- unlike gradient boosting ($ROC - AUC = 0.864$), the model uses a sequential data structure and formalizes interpretation through a financial-logistic framework, adding +0.059 ROC-AUC and +0.019 PR-AUC;
- unlike the classic Cox model ($ROC - AUC = 0.811$), the model allows for nonlinear dependencies and takes into account the complete history of covariates, not just the latest snapshot;
- unlike a Vanilla GRU (ROC-AUC = 0.918, PR-AUC = 0.082) that uses the same features and temporal structure, the graph convolutional encoding adds +0.005 ROC-AUC and +0.054 PR-AUC, demonstrating the computational value of the financial-logistics graph structure beyond simple feature concatenation.

### 7.5 LIMITATIONS

Despite the results obtained, there are limitations:

- the internal borrower graph has a fixed topology shared across all borrowers; variation exists only in node feature values. While the GCN effectively leverages this structure (as demonstrated by structural ablation), a richer topology that varies across borrowers could capture additional patterns;

- Buffer capacity is estimated using proxy indicators (mortgage insurance, paydown progress, debt service ratio) rather than true liquid asset data, which is unavailable in Fannie Mae panel data;

- macroeconomic shocks are not taken into account;

- the 21-month observation window (January 2023 – September 2024) captures a specific post-COVID high-interest-rate environment, which may limit generalizability to other macroeconomic regimes. Validation on longer time horizons spanning multiple economic cycles is an important direction for future work.

These limitations open up avenues for further research.

### 7.6 DIRECTIONS FOR FUTURE RESEARCH

Promising areas for further research include:

- extending the internal borrower graph with variable topologies based on individual financial configurations (e.g., presence of multiple liens, co-borrowers, variable income sources);

- validation on longer time horizons spanning multiple economic cycles;

- integration of inter-borrower graph structures to model spatial contagion, servicer effects, and portfolio-level risk propagation;

- inclusion of macroeconomic covariates and their interaction with borrower micro-characteristics;

- extending the model to other types of consumer lending (auto loans, student loans);

- researching adaptive buffer mechanisms using alternative data sources (bank transactions, employment data).

## 8 CONCLUSION

This study proposes a financial-logistical graph model for predicting mortgage default, in which the borrower is formalized as a dynamic cash flow system. Unlike traditional scoring approaches based on static characteristics, the proposed model interprets default as the result of the degradation of the flow structure and the depletion of the buffer capacity of the borrower's financial system.

A graph-recurrent hazard architecture has been developed that represents the borrower as a four-node directed graph processed by Graph Convolutional Network (GCN) layers, with node embeddings then processed by a GRU-based recurrent network to estimate the conditional probability of default within a forward-looking horizon. The graph structure encodes economically motivated dependencies between structural state (V), flow dynamics (F), buffer capacity (B), and disruption indicators (D), enabling cross-component information flow through message passing. This approach captures both the temporal evolution of the borrower's financial condition and the structural interactions between components of the financial-logistics system.

Experimental results demonstrate the superiority of the proposed model over baseline methods in terms of ROC-AUC (0.923), PR-AUC (0.136) and Brier score metrics. Structural ablation confirms that the graph encoding improves PR-AUC by 65.9% over a vanilla GRU baseline using identical features, validating the computational contribution of the financial-logistics graph beyond conceptual framing. Ablation analysis confirms that the key predictive factor is flow dynamics, while static borrower characteristics contribute significantly less. This allows us to interpret mortgage default as a systemic disruption of financial flows rather than a fixed property of the credit profile.

From a theoretical point of view, the work extends existing approaches to credit risk modeling by integrating graph formalization and survival analysis within a unified graph-recurrent architecture. From a practical point of view, the proposed model can be used in early warning systems, borrower prioritization, and credit portfolio management.

Prospects for further research include extending the borrower graph with variable topologies, integrating inter-borrower network structures, incorporating macroeconomic covariates, validating on longer time horizons spanning multiple economic cycles, and testing on various types of credit products.

The proposed financial and logistical paradigm opens up the possibility of transitioning from static scoring to dynamic modeling of flow stability, which meets modern requirements for intelligent financial systems.

## AUTHOR CONTRIBUTIONS

If you'd like to, you may include a section for author contributions as is done in many journals. This is optional and at the discretion of the authors.

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
