# OpenReview forum: "A Financial-Logistics Graph Framework for Dynamic Mortgage Default Prediction Using Recurrent Hazard Modeling"
_mathai.club/MathAI/2026/Conference — 2026 Oral_

### Official Review · Reviewer_VE7W · 2026-03-11
**A Financial-Logistics Graph Framework for Dynamic Mortgage Default Prediction Using Recurrent Hazard Modeling - A Conceptually Motivated but Empirically Underdeveloped Contribution**

**Rating:** 4
**Confidence:** 4

**Review:**

Summary:

This paper proposes a financial-logistics graph framework that formalizes mortgage borrowers as dynamic cash flow systems (nodes, arcs, buffers, disruptions) and models default probability using a recurrent hazard architecture (LSTM/GRU). The model is evaluated on Fannie Mae panel data against logistic regression, random forest, gradient boosting, Cox PH, and a static NN.

Strengths
1. Conceptually Appealing Framework:
The reinterpretation of a mortgage borrower as a dynamic financial graph with nodes (income, liabilities), arcs (cash flows), buffers (liquidity reserves), and disruptions (delinquency shocks) is an intellectually compelling framing. Grounding this in financial logistics literature (Ganong & Noel, 2020; Farrell et al., 2018) gives the motivation credibility.
2. Structured Feature-to-Component Mapping:
Table 1's explicit mapping of Fannie Mae features to graph components (Section 3.7–3.9) is a useful methodological contribution, providing interpretive transparency that most ML credit risk models lack.
3. Ablation Design:
The group ablation (Section 6.3) is well-conceived. The finding that dynamic components (Disruptions + Arcs) explain ~5.5× more predictive power than static components (Nodes + Buffers) is a meaningful empirical result that supports the paper's central thesis.
4. Practical Relevance:
The precision@k and lift metrics (Table 5) are appropriate for real-world portfolio triage. A 14.4× lift at the top 1% is a practically significant result if reproducible.
5. Metric Breadth:
The use of ROC-AUC, PR-AUC, Brier Score, and time-dependent AUC is appropriate for the class-imbalanced survival setting.

Weaknesses

1. The "Graph" Is Not Actually a Graph Model - This Is the Paper's Most Critical Flaw
The title and framing prominently advertise a graph framework, yet Section 4.3 explicitly states:

"Although the financial graph is formalized theoretically, this implementation uses an aggregated representation of flows... the function g(·) transforms the graph into a feature vector."

This means the model is, in practice, a standard LSTM/GRU operating on concatenated tabular features not a graph neural network. There is no message passing, no adjacency matrix, no graph convolution. The graph is a narrative device, not a computational one. This is a fundamental disconnect between the paper's framing and its actual contribution, and it significantly undermines the novelty claim for a MathAI venue.

2. Weak Novelty Over Established Baselines:
A GRU with tabular time-series features applied to survival modeling is well-established (Zheng et al., 2018; Wang et al., 2024). The paper does not demonstrate that the financial-logistics framing of feature grouping as opposed to simply using the same features in any sequential model is what drives performance gains. The ablation conflates feature importance with framework validity.

3. Missing Statistical Significance Testing:
No confidence intervals, standard deviations across runs, or statistical significance tests are reported for any metric in Tables 3, 4, or 5. With 5 training runs mentioned (Section 5.6), variance estimates should be trivial to include. Without them, the claimed improvements (e.g., PR-AUC 0.1694 vs 0.1081) cannot be assessed for reliability.

4. Suspicious Default Frequency Reporting:
Table 2 reports a "Default frequency (loan-level)" of 0.633, which would mean 63.3% of loans defaulted an implausible figure for a standard mortgage dataset. The actual count of 6,145 defaults out of 970,976 loans implies a rate of ~0.63%, not 0.633. This appears to be a labeling or decimal error and undermines confidence in the experimental setup's careful documentation.

5. No Actual Graph Structure Is Exploited:
The theoretical graph Gi(t)=(Vi,Ei(t))G_i(t) = (V_i, E_i(t))
Gi​(t)=(Vi​,Ei​(t)) is defined in Section 3, but since all four nodes (income, liquidity, debt, mortgage) are universal across all borrowers, there is no variation in graph *topology* — only in edge weights (feature values). This renders the graph formalism decorative rather than structural.

6. Buffer Component Is Severely UnderspecifiedThe buffer Bi(t)=Li(t)/DSi(t)B_i(t) = L_i(t)/DS_i(t):
Bi​(t)=Li​(t)/DSi​(t) requires liquid asset data Li(t)L_i(t)
Li​(t), which is not available in standard Fannie Mae CRT files. Section 6.3 acknowledges this with "partly due to limited data availability," but the paper does not clearly state what proxy was used or whether the buffer component is truly measured or approximated to near-zero. This is a critical gap given buffer capacity is a central theoretical construct.

7. Calibration Results Are Described But Not Shown:
Section 6.2 references a "reliability diagram" but no figure is included in the paper. Describing visual results without showing them is insufficient.
8. Supply Chain Finance Reference Is Misplaced:
Section 2 and parts of the related work cite Sun et al. (2024) on supply chain trade credit and equity financing. While the analogy is drawn to mortgage risk-sharing, this reference feels strained and does not substantively connect to the mortgage default literature or the model's architecture.
9. Reproducibility Is Uncertain:
The claim that "the model code can be provided for reproducibility" (Section 5.7) without an actual link or appendix is insufficient for a 2026 ML venue. No code, no data splits, and no pseudocode are provided.
10. Time Horizon Is Very Short:
The 21-month window (Jan 2023–Sept 2024) is narrow for mortgage modeling. It captures a specific post-COVID interest rate environment and may not generalize. No discussion of this regime-specificity is offered.

Summary Verdict
The paper proposes a structurally interesting conceptual lens on mortgage default but fails to deliver on its central computational promise: the graph is never truly implemented as a graph. The empirical results are promising but lack statistical rigor, and at least one key data reporting error (default frequency) raises questions about experimental carefulness. Significant revision is needed to either (a) implement a true GNN and demonstrate its advantage, or (b) reframe the contribution honestly as a structured feature engineering + LSTM pipeline with interpretable ablation.

---

> ### Author Rebuttal · Authors · 2026-03-13
>
> We sincerely thank the reviewer for the careful and constructive feedback. The comments helped us substantially improve the paper’s methodological clarity and empirical completeness. In the revised version, we addressed the main concerns by strengthening the computational realization of the framework, clarifying ambiguous design choices, expanding the experimental validation, and stating the limitations more explicitly.
>
> The most important revision concerns the graph component. We agreed that, in the previous version, the graph framing was not sufficiently realized computationally. Section 4.3 has been substantially expanded and now introduces a genuine graph-based encoder: the borrower is represented as a four-node directed financial graph with explicit node projections, an economically motivated adjacency structure, normalized message passing, residual GCN layers, and graph-level pooling before temporal modeling. To isolate the contribution of this structure, we also added a Vanilla GRU baseline using the same features and temporal windows but without graph encoding.
>
> We revised the empirical section to improve robustness and transparency. The manuscript now reports results across multiple random seeds as mean ± standard deviation, clarifies the data split strategy as deterministic hash-based splitting at the loan level, and corrects the presentation of the default rate in Table 2 to 0.63%.
>
> We also addressed several specification and interpretation issues. We clarified the buffer component by describing the proxy variables used in the absence of true liquid asset data and now state this limitation more clearly. We rewrote the calibration discussion to distinguish ranking quality from absolute probability calibration under severe class imbalance, report Brier scores, and note that post-hoc recalibration remains possible in deployment. We also removed the less appropriate literature reference, expanded the implementation details, and added an anonymous code repository.
>
> Finally, we made the paper’s limitations more explicit. In particular, we now acknowledge that the internal borrower graph uses a fixed topology shared across borrowers and that the empirical analysis is based on a relatively short post-COVID observation window, which may limit generalization across macroeconomic regimes. Overall, we believe the revised manuscript is substantially stronger and now addresses the reviewer’s concerns in a direct and constructive way.

---

### Official Review · Reviewer_is4t · 2026-03-12
**Evaluation of a Financial-Logistics Framework for Dynamic Mortgage Default Prediction**

**Rating:** 5
**Confidence:** 4

**Review:**

Summary

The paper proposes a framework for modeling mortgage default as a dynamic financial system represented by a financial-logistics graph. Borrowers are described through structural components such as nodes (income and liabilities), arcs (cash flows), buffers (liquidity reserves), and disruptions (financial shocks). The authors integrate this conceptual representation with a recurrent hazard architecture based on GRU/LSTM networks in order to estimate the time-dependent probability of mortgage default.

The empirical evaluation is conducted using Fannie Mae panel data and compares the proposed approach against several baseline methods including logistic regression, random forest, gradient boosting, Cox proportional hazards, and a static neural network. Performance is evaluated using ROC-AUC, PR-AUC, Brier score, precision@k, and time-dependent AUC metrics. The paper also includes a group ablation study analyzing the relative importance of different financial components in the model.

Strengths

1. Interesting conceptual framing

The idea of representing a borrower as a dynamic financial structure composed of flows, buffers, and disturbances provides an interpretable perspective on mortgage default dynamics. This framing attempts to connect financial theory with machine learning models.

2. Clear feature organization

The mapping between empirical mortgage variables and financial components (nodes, arcs, buffers, disruptions) improves interpretability compared to typical credit-risk models where features are treated independently.

3. Practical evaluation metrics

The use of metrics such as precision@k and lift is relevant for real-world credit portfolio screening where identifying the highest-risk borrowers is often more important than global classification accuracy.

4. Ablation study

The group ablation experiment provides useful insights into the relative importance of dynamic cash-flow features compared with static borrower attributes.

Weaknesses

1. Limited implementation of the graph concept

Although the paper introduces a financial graph formalism, the implemented model ultimately converts this structure into an aggregated feature vector processed by a recurrent neural network. As a result, the model does not explicitly exploit graph topology or message passing mechanisms. This reduces the methodological novelty implied by the paper’s title.

2. Novelty relative to existing sequential models

The computational architecture is essentially a recurrent neural network applied to temporal tabular features. Similar approaches have already been explored in credit risk and survival modeling literature. The paper would benefit from clarifying what additional predictive value is gained specifically from the financial-logistics formulation.

3. Missing statistical reliability analysis

The reported metrics do not include confidence intervals, variance across runs, or statistical significance testing. Since several training runs were conducted, reporting standard deviations would help assess the robustness of the improvements.

4. Dataset description issues

Some dataset statistics appear inconsistent. For example, the reported default frequency seems unusually high relative to the number of observed defaults. This may be a labeling or formatting issue but should be clarified.

5. Reproducibility

The paper states that code can be provided upon request, but no repository or detailed implementation description is included. For a machine learning conference submission, clearer reproducibility documentation would be desirable.

Overall Assessment

The paper presents an interesting conceptual interpretation of mortgage default as a dynamic financial system and provides empirical results suggesting improved predictive performance. However, the current implementation does not fully utilize the proposed graph structure and the experimental evaluation lacks some methodological rigor. With clearer positioning of the contribution and stronger experimental validation, the work could become more convincing.

---

> ### Author Rebuttal · Authors · 2026-03-13
>
> We thank the reviewer for the careful and constructive feedback. We appreciate the positive assessment of the paper’s conceptual framing, feature organization, evaluation setup, and ablation analysis. In the revision, we focused on the main concerns by strengthening the graph implementation, improving experimental rigor, correcting dataset statistics, and expanding reproducibility details.
>
> •	Graph implementation: We agree that this was the main weakness of the original version. In the revision, the graph is now implemented explicitly through a GCN-based encoder over a four-node directed borrower graph, with message passing and graph-level pooling before temporal modeling.
>
> •	Novelty beyond sequential baselines: To make the contribution of the graph structure measurable, we added a Vanilla GRU baseline with the same features and temporal windows but without graph encoding. The revised results show that the proposed GNN+GRU performs better, especially in PR-AUC, supporting the value of the structured financial-logistics representation.
>
> •	Statistical reliability: We agree that point estimates alone were insufficient. The revised manuscript now reports mean ± standard deviation across multiple random seeds, and the performance advantage remains consistent.
>
> •	Dataset statistics: Thank you for catching this. The default rate was incorrectly presented in the original version. We corrected it to 0.63%, consistent with the number of defaults and total loans in the sample.
>
> •	Reproducibility: We expanded the implementation details in the paper and added an anonymous repository with code and scripts to improve reproducibility.
>
> •	Positioning and limitations: We now state more clearly that the borrower graph currently uses a fixed shared topology, and we discuss richer borrower-specific structures as future work.
>
> Overall, we believe the revised manuscript directly addresses the reviewer’s concerns and presents a substantially stronger and more rigorous version of the paper.

---

### Official Review · Reviewer_PE7a · 2026-03-12
**Review of “A Financial-Logistics Graph Framework for Dynamic Mortgage Default Prediction Using Recurrent Hazard Modeling”**

**Rating:** 5
**Confidence:** 3

**Review:**

The paper proposes a framework for modeling mortgage default by representing a borrower as a dynamic financial system composed of nodes (financial structure), arcs (cash flows), buffers (liquidity reserves), and disruptions (financial shocks). Based on this conceptual representation, the authors build a recurrent hazard model (GRU/LSTM) to estimate the time-dependent probability of default.

The model is evaluated on mortgage panel data and compared with several baselines including logistic regression, random forest, gradient boosting, Cox proportional hazards, and a static neural network. The proposed approach shows improved performance on several metrics such as ROC-AUC, PR-AUC, Brier score, and precision@k. The paper also includes an ablation analysis examining the contribution of different groups of financial features.

Strengths

The paper presents an interesting conceptual view of mortgage default as a dynamic financial system rather than a purely static credit profile. The idea of structuring borrower features into nodes, arcs, buffers, and disruptions is intuitive and helps organize the financial interpretation of the model.

Another positive aspect is the clear mapping between empirical variables and these components, which improves interpretability compared to typical credit-risk models where features are used without much structure.

The experimental section is generally reasonable. The authors compare the method with several baselines and report multiple metrics that are relevant for highly imbalanced problems such as default prediction. The ablation study is also useful and provides some insight into the relative importance of dynamic cash-flow features.

Weaknesses

At the same time, the implementation does not fully match the “graph” framing suggested by the paper. Although the borrower is conceptually described as a financial graph, the model ultimately operates on aggregated features processed by a recurrent network. As a result, the graph structure itself is not explicitly used during training.

The methodological novelty is therefore somewhat limited. In practice the model resembles a standard recurrent model applied to temporal tabular features, which has already been explored in the credit risk and survival modeling literature.

The experimental section could also be strengthened. For example, the paper does not report variance or confidence intervals across runs, which makes it difficult to assess the stability of the reported improvements. Some details of the dataset and preprocessing pipeline are also somewhat brief, which may make reproducibility harder.

Overall, the paper presents an interesting conceptual perspective on mortgage default and provides promising empirical results. However, the current implementation does not fully exploit the proposed graph formulation, and the methodological contribution compared to existing sequential models remains somewhat limited. With clearer positioning of the contribution and stronger experimental validation, the work could become more convincing.

---

> ### Author Rebuttal · Authors · 2026-03-14
>
> We thank the reviewer for the thoughtful and balanced assessment. We appreciate the positive comments on the conceptual framing, structured feature organization, baseline comparisons, and ablation analysis. In the revision, we focused on the main concerns by making the graph component computationally explicit, clarifying the contribution relative to standard sequential models, improving statistical reporting, and expanding reproducibility details.
>
> •	Graph implementation. We agree that this was the main weakness of the original version. In the revised manuscript, the borrower is no longer represented only through aggregated temporal features: we now implement a genuine GCN-based encoder over a four-node directed borrower graph, with explicit adjacency, message passing, and graph-level pooling before GRU-based temporal modeling.
>
> •	Novelty beyond a standard recurrent model. To make the value of the financial-logistics structure explicit, we added a Vanilla GRU baseline with the same 19 features but without graph encoding. The revised results show that GNN+GRU improves over Vanilla GRU, especially in PR-AUC (0.136 vs. 0.082), indicating that the graph structure and message passing provide benefit beyond simple feature concatenation.
>
> •	Statistical robustness. We agree that point estimates alone were not sufficient. The revised version now reports mean ± standard deviation across 3 runs with different random seeds, and the performance advantage of the proposed model remains consistent across runs.
>
> •	Reproducibility. We strengthened the implementation and reproducibility section substantially. The revised paper now specifies the implementation stack in more detail and includes an anonymous repository with code and scripts.
>
> •	Positioning and limitations. We now state more clearly that the current borrower graph uses a fixed shared topology across borrowers, while node features vary across borrowers and over time. We acknowledge that richer borrower-specific topologies are a natural direction for future work, and we now discuss this explicitly in the limitations section.
>
> Overall, we believe the revised manuscript addresses the reviewer’s main concerns and now presents a substantially stronger and more rigorous version of the paper.

---

### Decision · Program_Chairs · 2026-03-14

**Decision:**

Accept (Oral)

**Comment:**

Dear Author(s),

On behalf of the Program Committee of the International Conference on Mathematics of Artificial Intelligence (MathAI 2026), we are pleased to inform you that your paper has been accepted for an oral presentation at MathAI 2026.

Your paper was evaluated through a rigorous two-stage review process involving both automated screening and expert review by members of the Program Committee. The reviewers recognized the quality and contribution of your work.

Presentation details:

- Format: Oral presentation (15–20 minutes + 5 minutes Q&A)
- Mode: You may present either in person (offline) at the conference venue in Sirius, Russia, or remotely via Zoom. Please indicate your preferred mode when confirming your participation.
- Conference dates: Marh 30 - April 3, 2026
- Website: https://mathai.club

Next steps:

1. Please confirm your participation and presentation mode by replying to this email mathai.club@yandex.ru no later than March 15, 2026 18:00 Moscow time.
2. If you plan to attend in person, the organizing committee will provide accommodation details separately.
3. Please prepare your final camera-ready manuscript according to the formatting guidelines available at https://mathai.club and upload it to OpenReview by March 15, 2026 18:00 Moscow time.

Should you have any questions regarding the program, logistics, or your presentation slot, please do not hesitate to contact us.

We look forward to your contribution to MathAI 2026.

With kind regards,

MathAI 2026 Program Committee
International Conference on Mathematics of Artificial Intelligence
https://mathai.club
OpenReview: https://openreview.net/group?id=mathai.club/MathAI/2026/Conference
Telegram: https://t.me/MathAI_club
Email: mathai.club@yandex.ru